# Determinants of primary healthcare providers' readiness for integration of ART services at departmental levels: A case study of Lira City and District, Uganda

**Emmanuel Asher Ikwara[1], Lakeri Nakero[2], Maxson Kenneth Anyolitho[2], Rogers Isabirye[3], Syliviah Namutebi[3], Godfrey Mwesiga[4], Sean Steven Puleh[1]\***

1 Department of Epidemiology and Biostatistics, Faculty of Public Health, Lira University, Lira City, Uganda,
2 Department of Community Health, Faculty of Public Health, Lira University, Lira City, Uganda,
3 Department of Midwifery, Faculty of Nursing and Midwifery, Lira University, Lira City, Uganda,
4 Department of Psychiatry, Faculty of Medicine, Lira University, Lira City, Uganda

\* spuleh@lirauni.ac.ug

**Data Availability Statement:** All relevant data are within the paper and its Supporting information files.

## Abstract

### Background

Decreasing or flattening funding for vertical HIV services means that new and innovative ways of providing care are necessary. This study aimed to assess the determinants of readiness for integration of Antiretroviral Therapy (ART) services at the departmental level among primary health care providers (PHCP) at selected health facilities in Lira District.

### Methods

A cross-sectional survey employing mixed methods approaches was conducted between January and February 2022 among 340 primary healthcare practitioners (PHCP) at selected health facilities in Lira district. An interviewer-administered questionnaire was used to collect quantitative data. Quantitative data was analyzed using Stata version 15. and presented as proportions, means, percentages, frequencies, and odds ratios. Logistic regression was used to determine associations of the factors with readiness for ART integration at a 95% level of significance. Thematic analysis was used to analyze qualitative data.

### Results

The majority 75.2% (95% CI; 0.703–0.795) of the respondents reported being ready for the integration of ART services. PHCPs who were aware of the integration of services and those who had worked in the same facility for at least 6 years had higher odds of readiness for integration of ART, compared with their counterparts [aOR = 7.36; 95% CI = 3.857–14.028, p-value <0.001] for knowledge and duration at the current facility [aOR = 2.92; 95% CI = 1.293–6.599, p-value < 0.05] respectively. From the qualitative data, the dominant view was that integration is a good thing that should be implemented immediately. However,

**Funding:** Research reported in this publication was supported by the Fogarty International Center (U.S. Department of State's Office of the U.S. Global AIDS Coordinator and Health Diplomacy [S/GAC] and the President's Emergency Plan for AIDS Relief [PEPFAR]) of the National Institutes of Health under Award Number R25TW011210. The content is solely the responsibility of the authors and does not necessarily represent the official views of the National Institutes of Health. The funders had no role in study design, data collection and analysis, decision to publish, or preparation of the manuscript.

**Competing interests:** The authors have declared that no competing interests exist.

several challenges were noted, key among which include limited staffing and drug supplies at the facilities, coupled with limited space.

## Conclusions

The study reveals a high level of readiness for the integration of ART services at departmental levels among Primary Healthcare Providers. Notably, PHCPs knowledgeable about integration and those who spent at least six years at the current health facility of work, were strong determinants for the integration of ART services in resource limited settings. In light of these findings, we recommend that policymakers prioritize the implementation of training programs aimed at upskilling healthcare workers. Furthermore, we advocate that a cluster randomized controlled trial be conducted, to evaluate the long-term effects of this integration on overall health outcomes.

## Introduction

Sub-Saharan Africa (SSA) bears the greatest burden of HIV/AIDS, accounting for the majority of new infections and deaths worldwide [1]. In Uganda, approximately 1.4 million people, both adults and children, are infected with HIV [2]. Many African countries, including Uganda, have adopted the UNAIDS 95-95-95 target of ensuring that 95% of people are aware of their HIV status, 95% of people diagnosed with the virus receive antiretroviral therapy, and 95% of those on therapies have undetectable viral load [3]. However, the UNAIDS-focused target of near-universal 95% antiretroviral therapy (ART) access for people living with HIV (PLHIV) by 2030 has several potential challenges [3].

The current success in the effectiveness of the vertical programs has been attributed to the donor funds [4]. In Uganda and across the region, HIV services are provided by vertically operating separately from other health system functions [5]. Despite the successes and achievements already realized, several questions around its sustainability are becoming very critical. Moreover, there is evidence of decreasing or flattening funding for vertical HIV services [6]. Additionally, many patients are still lost at various stages of the continuum of care [7]. This, therefore, implies that new and innovative ways of providing care are vital. A possible approach that has shown positive outcomes in improving HIV/AIDS services along the continuum of care could be horizontal integration at the point of service delivery in the various departments [8]. Several actors have also highlighted the need for the integration of ART management services at various health facility department levels [9]. Furthermore, evidence of the medical and public health benefits of HIV/AIDS service integration seems to back it up [10]. Integration in the context of this study refers to the act of combining ART services at departmental levels with other non-HIV-specific services, such as out-patient departments (OPD), primary healthcare (PHC), in-patient departments (IPD), maternal, newborn, and child health (MNCH) services, sexual and reproductive health (SRH), and family planning (FP), among others.

Some of the determinants can be classified into, the health systems factors (availability of resources, funding, personnel, and infrastructure), provider factors (knowledge, skills, and attitudes of healthcare providers towards ART services, such as their ability to provide counseling and support), community factors (social and cultural norms, stigma, and discrimination) can also influence the integration of ART services in health facilities [11–14]. Analysis of these determinants will enable health systems to identify gaps and challenges that need to be

addressed to effectively integrate ART services. The benefits of integration include lowered costs at primary health clinics [15], improvement in HIV service uptake, health outcomes, as well as outcomes related to other services [8]. In addition, this novel idea would benefit patients with comorbidities in terms of conformity of care and increased access to HIV/AIDS services. It can also allow healthcare providers to share the workload for all patients, resulting in more efficient use of resources and reduced patient waiting time [16]. Available evidence also suggests that integration could reduce discrimination by "normalizing" HIV services [17].

When viewed from the providers' and funders' perspective, integration has the potential to improve processes and resource allocation [9, 18]. It is also important that people get the care they need, when they need it, in ways that are user-friendly, thereby achieving desired results and value for money. Currently, there is an inadequate understanding of healthcare workers' readiness for the integration of ART services at the departmental level. This could pose challenges to its implementation. This study, therefore, assessed the determinants of primary healthcare providers' readiness toward the integration of ART management services at departmental levels in health facilities in Lira district, to inform policy formulations and guide successful implementation.

## Materials and methods

### Design

The study adopted a descriptive cross-sectional study design and was conducted using mixed methods approaches to data collection and analysis. The qualitative method enabled us to document the achievements and challenges faced by the primary healthcare workers in the provision of ART services at the selected health facilities. The quantitative objectives were to assess the level of readiness of the health facilities regarding the integration of ART services and their determining factors. The choice of the design was informed by the fact that a comprehensive understanding of the readiness would guide policy formulation. Furthermore, a proper understanding of the achievements and challenges faced by the health facilities would guide the directions for future service provision.

### Study setting and population

The study was conducted in Lira District, located about 340 kilometres north of Kampala, Uganda's capital city. Four health facilities, comprising Lira Regional Referral Hospital (LRRH), PAG mission hospital, Ogur Health Centre IV, and Amach Health Center IV, were selected for the study. LRRH is the major referral hospital in the sub-region (Lango) with a total bed capacity of 254 running different units as per the government health systems structure. PAG Mission Hospital is a faith-based, private, not-for-profit institution registered with the Ugandan MoH at level five. Meanwhile, both Ogur and Amach are public health center IVs administered by the MoH with OPD, IPD, a theatre, maternity services, and medical departments. The study population consisted exclusively of primary healthcare providers (PHCP's) from the above-selected facilities. The PHCPs involved in the study included medical doctors, pharmacists, clinicians, nurses, midwives, laboratory technicians, and counsellors. Health workers who were not full-time employees of the participating health facilities were excluded.

### Sample size and sampling criteria

We purposefully selected the four health facilities in the greater Lira district, considering their number of departments. A total of 340 PHCP who were available during the interview days

were recruited for the quantitative survey using a census. We also purposively recruited PHC providers for key informant interviews depending on their roles within the departmental units and work experience (at least 6 months at work). We structured focus group discussions consisting of multi-disciplinary PHC workers.

## Data collection and analysis

Data was collected in January and February 2022. The study tools for interviews were modified from the WHO reproductive health readiness assessment-hexagon tool [19]. Readiness was measured as a binary outcome. According to this tool, most of the determinants were in the domains of readiness assessment, including needs, fit, resources, capacity, readiness, and evidence. The tool was modified and designed by the research team to fit the context of the study setting. The modified tool was pretested before actual data collection and the information obtained was used to improve the tool. A total of 20 key informant interviews (KII) and four focused group discussions (FDG) were conducted at the four selected health facilities by three experienced interviewers of social science background.

The quantitative data collected was checked for completeness and later entered into SPSS version 23 with consistency checks to ensure correctness. The dataset was cleaned for out-of-range values and exported to STATA 15 (StataCorp, College Station, TX) software for analysis. We conducted a descriptive analysis to determine the proportions of the different variables in the respondents' characteristics, such as place of residence, age, sex, and marital status, among others. At the bivariate level, chi-square tests were performed to determine the association between dependent and independent variables. Further, odds ratio analyses were used to compute the unadjusted associations between the use of HIV prevention strategies and independent variables, including socio-demographic characteristics (such as age, marital status, level of education, sex, and occupation of participants). The results were expressed in terms of odds ratio with a 95% level of confidence and a P-value $< 0.05$. Finally, variables that were significant in the bivariate analysis ($p < 0.2$) were considered for the multivariable analysis. Logistic regression was performed to come up with a suitable model to explain the determinants of the PHCP readiness for ART integration at the departmental level and the statistical significance of $p<0.05$. Qualitative data collected was transcribed and entered into Nvivo version 12 software ready for onward analysis. A seven-step thematic analysis model by Clerke and Braun was used to analyse qualitative data using thematic analysis [20]. The steps include 1) transcription, 2) reading and familiarization 3) coding, 4) searching for themes, 5) review of themes, 6) naming the themes, 7) finalizing the analysis and interpretation of the results. A total of 15 themes and subthemes were developed from the data which include among others: Capacity and lack of capacity, qualification, experience, staffing, preparedness and unpreparedness, knowledge gap, unique nature of ART services, Need, fit and evidence. To ensure rigor and trustworthiness of the data the research team employed member checking, triangulation, detailed transcription, and following systematic plans and coding [21].

## Ethical consideration

The study protocol was reviewed and cleared by the Gulu University Research and Ethics Committee (GUREC-2021-173). Approval to conduct the study from Uganda was sought from the Uganda National Council for Science and Technology (UNCST). Administrative permission was obtained from the Resident City Commissioner of Lira City and the Resident District Commissioner of Lira District, the Chief Administrative Officer and the District Health Officer. Further permission was obtained from the heads of the selected health facilities, and written informed consent was sought from all respondents before interviews commenced.

## Results

### Socio-demographic characteristics of participants

A total of 340 participants were interviewed using an interviewer-administered questionnaire. Nearly 70% of respondents were less than 35 years old, and a large proportion of them, 65.6% (223/340), were employed by the government of Uganda. The mean age of the respondents was 33.2 years (standard deviation = 8.8), with the majority of 55.3% (188/340) being males. The majority of respondents, 84.4% (287/340), were aware of service integration, and slightly more than half (174/340) had worked at the current health facility for 2 to 5 years. A significant proportion of the respondents 75% (255/340) were trained in ART management, 67.9% (231/340) are currently engaged in ART management and 83.2% (283/340) reported that the implementation team had the capacity and was ready for wider implementation of integration (Table 1).

### Readiness by the healthcare providers regarding the integration of ART services

From the study, majority 75.2% (255/340), of the respondents reported being ready for the integration of ART services at the departmental level. Using the domains of readiness, nearly universal 94.4% (321/340) of the healthcare providers interviewed reported that integration meets the need for HIV management, and 81.8% (278/340) felt it fits the current guidelines by the ministry of health. A huge proportion, 79.1% (269/340) of the study participants noted their health facilities were prepared (Table 2).

**The qualitative aspects of readiness.** From the qualitative point of view, participants were asked to state their level of readiness. Key themes included the capacity of the facility; preparedness; fitting the current policy; evidence; resources; needs; and reasons for the views. Whereas respondents from the quantitative side generally reported being ready for integration, participants from both key informant interviews and focus group discussions expressed positivity with some reservations regarding the same. For instance, in this study, we assessed healthcare facilities' readiness for ART services integration.

**The capacity to handle ART integration.** Participants were asked to share their views on health facilities' capacity to handle ART integration. According to their responses, participants understood capacity in terms of qualifications, experience, and staff.

*Qualification.* In their own opinion, generally, healthcare facilities have staff who are qualified to handle ART services. Participants mentioned that they have staff with degrees, diplomas, and certificates in various fields suitable for the services.

*"The qualification of the human resource within the ART clinic is very okay, we have the doctors and nurses; we have staffed with degrees, diplomas, and certificates. Peer mothers who help us to follow up with clients in the communities"*

*(FGD participant 1 Amach HCIV)*

*"Homan resources are qualified, if they are to be interested they would because you know ART is all about the interest. The qualification is okay, we have the doctors, senior medical officers, and medical officers, and we have nurses.*

*(KII 3 Amach HCIV)*

*Experience.* In addition, the staff are experienced enough, with some having worked in ART sections already. Besides, they said that the number of staff at the moment is sufficient for a

**Table 1. Socio-demographic characteristics of respondents.**

| Variables | Frequency (n = 340) | Percentage (%) | p-value |
|---|---|---|---|
| **Sex of respondents** | | | |
| Females | 152 | 44.7 | 0.720 |
| Males | 188 | 55.3 | |
| **Age of respondents (years)[a]** | | | |
| 20–35 years | 235 | 69.9 | 0.585 |
| ≥36 Years | 101 | 30.1 | |
| **Marital status** | | | |
| Single | 117 | 34.4 | 0.034 |
| Married | 223 | 65.6 | |
| **Employer of the health worker** | | | |
| Government | 223 | 65.6 | 0.598 |
| Private not for Profit | 117 | 34.4 | |
| **Cadre of the health worker [b]** | | | |
| Nurse | 152 | 45.2 | 0.231 |
| Midwife | 59 | 17.6 | |
| Counsellor | 27 | 8.0 | |
| Lab Technicians | 44 | 13.1 | |
| Clinical Officers | 16 | 4.8 | |
| Pharmacy Tech | 17 | 5.1 | |
| Medical Officer | 21 | 6.3 | |
| **Duration at the health facility** | | | |
| ≤1 Years | 57 | 16.8 | 0.005 |
| 2–5 Years | 174 | 51.2 | |
| ≥ 6 Years | 109 | 32.1 | |
| **Are you aware of the integration of services at the departmental level** | | | |
| No | 53 | 15.6 | < 0.001 |
| Yes | 287 | 84.4 | |
| **Are you currently engaged in ART management** | | | |
| No | 109 | 32.1 | < 0.01 |
| Yes | 231 | 67.9 | |
| **Health worker was trained on ART management** | | | |
| No | 85 | 25.0 | 0.899 |
| Yes | 255 | 75.0 | |
| **Capacity of the implementation team and readiness for wider implementation of integration** | | | |
| No | 57 | 16.8 | 0.376 |
| Yes | 283 | 83.2 | |
| **Level of the health facility** | | | |
| Health center IV | 57 | 16.8 | 0.774 |
| PNFP level V hospital | 117 | 34.4 | |
| Regional Referral Hospital | 166 | 48.8 | |
| **Location of the health facility** | | | |
| Rural | 57 | 16.8 | 0.528 |
| Urban | 283 | 83.2 | |

[a] and [b] some missing values

**Table 2. Domains of readiness assessment among the healthcare providers.**

| Variables | Frequency (n = 340) | Percentage (%) |
|---|---|---|
| **Integration meets the need for the HIV management** | | |
| No | 19 | 5.6 |
| Yes | 321 | 94.4 |
| **Integration of ART services fit the current guideline** | | |
| No | 62 | 18.2 |
| Yes | 278 | 81.8 |
| **Health facility is prepared for integration** | | |
| No | 71 | 20.9 |
| Yes | 269 | 79.1 |
| **Health system capacity for implementation at all levels to improve and sustain ART** | | |
| No | 62 | 18.2 |
| Yes | 278 | 81.8 |
| **The available system and resources at this facility suitable for integration** | | |
| No | 69 | 20.3 |
| Yes | 270 | 79.7 |
| **Evidence integration can improve the outcome of treatment among HIV-positive clients** | | |
| No | 19 | 5.6 |
| Yes | 321 | 94.4 |

start. Maybe with time, more staff can be added. And with qualifications, continuous medical education would help equip those who don't know and refresh those who had the training earlier. Below is what some of the participants had to say in their own words:

> *"With ART is about the experience, even the psychiatric nursing officer can handle, we have the senior nursing officer, enrolled nurses"*
>
> *(KII 3 Amach HCIV)*
>
> *"Yes, because here we have three clinical officers in this facility. Yes, then two medical officers, we have records, so everything is in place and I believe the facility can handle that integration"*
>
> *(KII 11 Ogur HCIV)*

*Lack of capacity to handle ART service integration.* Although there was a general feeling that the facilities had sufficient capacity to handle integration, a few participants had the opposite opinion. This was explained in terms of knowledge gap to handle ART services and unique nature of ART services respectively.

*Knowledge gap.* Such participants reasoned that there might exist knowledge gaps among staff. This according to them is because not everyone is trained in HIV-related matters, which would pose a serious risk of increased mortality. In addition, those who held this view argued that even for those who claim to be skilled and knowledgeable in HIV matters, the knowledge keeps changing because new things keep emerging. Some of them had the following to say in support of their views:

> *"About the capacity, the staff are well qualified but only that some of them lack knowledge about certain services in some departments, like TB ward where it is very hard for someone working maternity to work there and vice versa"*
>
> *(KII 8 Amach HCIV)*

*Unique nature of ART services.* Other participants argued that despite their qualifications, ART services are unique which some staff have not been trained. That is why they lack the knowledge to handle.

> *"Haa. . . people are qualified but you know HIV is a unique thing, first of all when you are at school, the kind of knowledge you get is quite different from what you get from the field, if this is going to be taken as a normal thing, it means we have to go into the curriculum that has to adapt and integrate this so that whoever is going to come out as a qualified medical staff would be able to handle HIV as any other condition"*

> *(FGD Participant 2 PAG Mission hospital)*

**Preparedness of the health facility for integration.** Concerning readiness for ART service integration in terms of preparedness, participants understood it to mean the presence or absence of infrastructure, equipment, andspace, among others. There were mixed opinions of equal measure by both those who said they were prepared and those who said the facilities were not prepared for integration.

*Availability of infrastructures, equipment and space.* According to those who said that the facilities were prepared, they gave reasons such as availability of infrastructure, presence of equipment and space, including the welcoming nature of the staff. Participants reasoned that they would be able to accommodate the increased number of clients due to available infrastructure. Furthermore, the space will allow them to serve the clients effectively. Below is what some of the participants said in support of their views:

> *"If the integration is to be in place, the government is also aware, I think with that I may say the hospital may be prepared because all the required equipment must have been in place with the support from the government"*

> *(KII 10 PAG Mission hospital)*

> *"I think for us in OPD here, since we have the next building, we are ready except we need that knowledge"*

> *(KII 12 Lira RRH)*

*Unpreparedness.* On the other hand, some participants who were opposed to preparedness argued that the facilities are not yet ready because of the following reasons:

> *"Inadequate drugs, Inadequate infrastructures like buildings, beds for clients, bleeding machines, radiation machines, etc. Knowledge gaps among the health workers about ART services; fear among clients if they are to be mixed with normal people"*

> *(FGD Participants 1,8, 5,3, Lira RRH)*

> *"Not yet ready: Because of inadequate staff that facility has, Inadequate building which is not enough for all the clients, knowledge gap among the few staffs available"*

> *(KII 15 PAG Mission hospital)*

As can be observed from the above responses, relatedly, both those who opposed and those who supported their preparedness gave similar but opposite viewpoints on the same. For instance, while some participants said they had the infrastructure, others said the

infrastructure was not enough. Same with knowledge, which was also mentioned under capacity.

**Integration meets the needs for HIV management.** In terms of need, participants were asked to express their views on whether they thought integration would meet the need for HIV management. As with the preparedness for ART service integration, participants gave varied but equally opposite opinions. There was an equally strong view that the integration of ART meets the need for HIV management just like those who are opposed. Those who stated that there was a need gave examples of the availability of record systems, computer equipment, and data management systems, among others. It is not clear, however, if these are sufficient to understand the need for integration.

**Integration fits the current guidelines.** In terms of fitness for integration, participants also gave varied views, both of which had equal positions, with some saying the protocols and guidelines for integration will fit the current system. Furthermore, the existing staff and their qualifications will make it fit the integration without any challenges. They said, in their own words, that. . .

*"Yes, it will fit and match, because it is going to expand knowledge into staffs and what we were already doing is not far from integrations so I believe the protocol and guidelines, we have doctors and nurses who will be working together"*

*(KII 10 PAG Mission hospital)*

*"The current guidelines talk about the integration of services in the community, and also the facility. It fits because of that linkage and referrals"*

*(KII 7 Lira RRH)*

*"The system will fit without any problem because it is the same professionals operating and some trained medical workers"*

*(KII 19 Ogur HCIV)*

*Unfit.* On the other hand, those who are opposed to fitness argue that the absence of guidelines and policies makes it difficult for integration to fit. The guidelines are supposed to explain how integration should be done, who should do what where, when and how. In the absence of this, it becomes difficult to operate. Interestingly, some of the reasons given against fitness were also the same as those mentioned earlier for preparedness and capacity. They include knowledge gaps, inadequate staff, limited infrastructure, and many others. In support of their views, some of the participants said that. . ..

*"All, I see this facility is not yet ready for integration since we still don't have the guidelines for how we should work, the infrastructures are still very poor, we even lack the staff to promote the integration, clients themselves need to be sensitized on the integration of ART clinic with other department and how it will help them assess their services"*

*(FGD participants, Lira RRH)*

*"We don't have a specific guideline; we are using the national guideline adopted by the ministry of health, including integration. It has not been taken up seriously, there is a need to bring staff on board"*

*(FGD Participants PAG Mission hospital).*

**Evidence integration can improve outcomes for HIV-positive clients.** As regards evidence that integration of ART services can improve the outcomes for HIV-positive clients, participants also gave different opinions and views. From the data, however, the most pronounced opinion is that there will be evidence of integration improving the lives of HIV-positive clients. This will be seen in various ways, such as increased adherence resulting from a reduction in stigma. This will also lead to a suppressed viral load and hence improvement in the health status of the positive clients. In addition, there will be evidence of an early diagnosis, reduced waiting time and ease of service, and generally improved ART service delivery. In their own words, the participants said thus:

*"One thing that will come over is the treatment, it will improve clients' satisfaction, load, it will reduce waiting time for clients, retention will also be good, viral suppression as well"*

*(KII 7 Lira RRH)*

*"In the maternity ward, all the positive mothers, and their children are bone negative, all the viral loads are being taken without problem because all the department is doing it, which has reduced the workload of ART services staff"*

*(KII 9 Amach HCIV)*

*"Management of ART patients will be easy, even men who have poor health-seeking habits will come"*

*(KII 5 PAG mission hospital)*

*"It will reduce patients waiting time. Increase in the number of young adults attending ART services"*

*(KII 2 Ogur HCIV)*

*No tangible evidence.* As noted earlier, a few of the healthcare workers still felt that there would be no tangible evidence of integration. In their opinion, they reasoned that integrating ART services would produce negative evidence such as a reduction in client turn-up, which would lead to their loss. This is due to fear of the stigma that will arise. Also, there will be evidence of wastage of ARV drugs, as some will not be taking them from the different units. In addition, it will cause discomfort for some of the clients who will not want to be known or seen by others. Below is what they had to say:

*"There will be lost clients, like when these people are mixed up there will be no way that you tell me that a patient will never get to realize that she is on ART services, because like in terms of dispensing, that is when I know that the mistake might come out"*

*(KII 20 Ogur HCIV)*

*"I don't know whether our patients will be comfortable here, I don't know whether they will go by waiting time, and yet the emergencies will also be coming in"*

*(KII 12 Lira RRH)*

*"I think this will discourage many clients to come. The number of clients will also reduce this is because of the loss of those clients, since some of them will go and never come back as a result of stigma"*

*(FGD participants, Lira RRH)*

*Inadequate resources.* Finally, our data shows that there was generally a dominant view that the resources available from the facilities are not enough to warrant ART service integration. Participants from across diverse units and facilities agreed that the facilities have inadequate human resources and infrastructure such as space, equipment, and machines. They also noted that even the existing staff are not fully trained to handle ART services, hence making them unready. Some of the participants said these things in their own words:

*"There is inadequate building to support the integration of the ART services. Health workers do not have enough skills to handle ART services, Insufficient drugs, Inadequate staffs both with knowledge gap and without knowledge gap about ART services management"*

*(FGD participants, Lira RRH)*

*"We need more resources as we said earlier like registers for proper documentation, human resource"*

*(FGD Participant 4 PAG Mission hospital)*

*"The resources are not enough for the ART clinic. Alright, the resources will not be good enough to welcome the integration of ART services into all departments"*

*(KII 20 Ogur HCIV)*

Few participants insisted that what the facilities have at the moment is sufficient, to begin with. Maybe a few more of them can be added as some of them had these to say;

*"Those things are there, maybe the computers which are mostly only with record keepers"*

*(KII 10 PAG Mission hospital).*

*"Resources are enough. We have the drugs which are enough, the services we are supposed to render to them even the laboratory personnel are there"*

*(KII 13 Amach HCIV)*

*"For now, services are suitable as you see the monitoring and evaluation bit of it. Ok resources like human resources is always a challenge, that section is even worse, sometimes they are not enough, space challenges, storage facilities, racks "*

*(KII 17 Lira RRH)*

As can be seen from the above, it appears that those who felt that the resources were insufficient to warrant integration are the ones who do the actual work, while those who said the resources were sufficient seem to be the ones in charge and heads of units who may not be in touch with reality.

## Health systems determinants of readiness of primary health care workers for integration of ART services at the departmental level

In this study, healthcare providers who were aware of integration services were associated with higher odds of readiness for integration of ART, compared with healthcare providers who were not aware of integration services. And the association was statistically significant [aOR = 7.36; 95% CI = 3.857–14.028, p-value <0.001]. Our study shows that knowledge is very critical in the provision of healthcare to clients, and those who are aware are likely to welcome the idea. The healthcare providers who had worked in the same facility for at least 6 years had

**Table 3. Determinants of healthcare providers' readiness for integration of ART.**

| Variables | Ready for integration of ART services | | | |
| --- | --- | --- | --- | --- |
| | Yes | No | Crude OR (95%CI) p-value | Adjusted OR 95% CI |
| **Health worker is aware of the integration of services at departmental levels** | | | | |
| No | 33 | 51 | 1.00 | 1.00 |
| Yes | 235 | 20 | 7.60 (4.039–14.312) *** | 7.36 (3.857–14.028) *** |
| **Health worker was trained on ART management** | | | | |
| No | 59 | 20 | 1.00 | |
| Yes | 196 | 64 | 1.04 (0.581–1.855)0.899 | |
| **Health worker is engaged in ART services** | | | | |
| No | 71 | 37 | 1.00 | |
| Yes | 184 | 47 | 2.04 (1.225–3.399) ** | |
| **Employer of the Health workers** | | | | |
| Government | 165 | 57 | 1.00 | |
| Private not for profit | 90 | 27 | 0.59 (0.354–0.990) * | |
| **Healthcare workers considered the health facility ready for integration** | | | | |
| No | 17 | 45 | 1.00 | |
| Yes | 238 | 39 | 1.15(0.681–1.947) 0.598 | |
| **Duration of the health worker at the current health facility** | | | | |
| ≤ 1 year | 36 | 21 | 1.00 | 1.00 |
| 2–5 years | 127 | 47 | 1.58(.836–2.971) 0.159 | 1.35 (0.674–2.688)0.401 |
| At least 6 years | 92 | 16 | 3.35(1.575–7.143) ** | 2.92 (1.293–6.599) * |
| **Location of the health facility** | | | | |
| Rural | 41 | 16 | 1.00 | |
| Urban | 214 | 68 | 1.23(0.648–2.327) 0.528 | |
| **Level of the health facility** | | | | |
| Health Center IV | 41 | 16 | 1.00 | |
| Health center V | 90 | 27 | 1.30 (0.633–2.673)0.048 | |
| Regional Referral Hospital | 124 | 41 | 1.18 (0.600–2.322)0.631 | |

Level of significance

*p<0.05,

**p<0.01,

***p<0.001

higher odds of readiness compared to those who had worked for less than 1 year. And the association was statistically significant [aOR = 2.92; 95% CI = 1.293–6.599, p-value = <0.05] (Table 3).

## Discussion

In resource-limited settings, integrating antiretroviral treatment (ART) services into departmental levels at healthcare facilities is an effective strategy for improving HIV care outcomes. To the best of our knowledge, this is the first study assessing the determinants of readiness of the primary healthcare providers for the integration of ART services at departmental levels in this setting. Healthcare provider readiness for integration reflects the disposition or ability of a healthcare provider, regardless of the cadre, to offer ART and other clinical services at a single point in time to make the most of available resources. In our study, readiness was

assessed as a composite (binary outcome) based on the WHO domains of readiness assessment, which included needs, fit, resources, capacity, readiness, and evidence.

Regarding the readiness of PHC providers for integration of ART services, results from our study show that a high proportion of PHC providers reported being ready for the integration of ART services into departmental levels. This finding is consistent with the evidence from a study in which approximately 78% of staff were ready for the integration of routine rapid HIV screening in urban family planning (FP) clinics [22]. This can be explained by the fact that in both studies have been done in a similar context. Therefore, integrating ART services will not be difficult because the majority of PHC providers have been trained and many have previously participated in integrated PHC services such as TB and MCH. In the event policymakers decide to integrate ART services, most healthcare providers would be willing to take up the policy change positively.

Furthermore, in our study, the majority of staff reported that integrating ART services at the departmental level would improve ART therapy outcomes among PLWHIV. Our findings are consistent with evidence from elsewhere, which found that integrating HIV services increased patient satisfaction, perceived quality of care, and patient access to services, and patient health outcomes [23, 24]. In addition, our findings indicate that integration may reduce stigma and discrimination, which is consistent with data from other studies suggesting that integration may reduce discrimination by 'normalizing' HIV services [17, 25, 26]. On the other hand, patient outcomes may also be better and costs lowered at primary health facilities [15] as a result of integration of ART services at departmental level. Therefore, the integration of HIV services is feasible and has the potential for positive outcomes to improve health and health systems [27]. However, according to one study, there was no statistically significant difference in viral suppression between integrated and separate services [28].

Our study also reveals some concerns by the respondents about integration increasing the workload among healthcare providers. This finding is consistent with evidence from studies that suggested integration had a risk of overburdening healthcare providers [5, 29], especially where the prevalence of HIV is very high [23]. This could be attributed to increased client turn-up as a result of receiving all necessary services from the same point of care, which means more work for the staff. Furthermore, other studies have found insufficient human resource capacity to provide additional services [30], which could be difficult for the already overburdened health system. A plausible reason could be related to the context in which the studies were conducted. Other evidence on the other hand seem to suggest that one way to address these challenging issues, is to integrate healthcare at all levels [31]. The concerned authorities should conduct massive recruitment and conduct training if the integration of ART at the departmental level becomes policy.

Training and capacity building are important for preparing PHC providers for the integration of ART services. In this study, our result show that the majority of the primary care providers were trained in ART management which is very important in the HIV care cascade. This is consistent with other studies' findings that competency-based capacity-building for various health worker cadres along the training continuum are effective [32]. Furthermore, for healthcare providers to provide multiple desired services, including ART, at a single point of contact, they must be adequately trained in all aspects [33]. It is worth noting that vertical HIV care had funding and numerous opportunities for staff training on HIV care [34]. Fortunately, most of the medical-related training institutions in Uganda provide HIV training as a cross-cutting issue something that would add value to ART integration.

Finally, according to the findings of this study, many participants believed that the resources available from the facilities were insufficient to justify the integration of ART services. Similar findings show that integrating services can result in more efficient use of

available resources, such as human resources, medical supplies, and drugs [23]. This means that integration is particularly important in resource-constrained settings where healthcare workers and medical supplies are scarce. Comparable to studies also noted that the success of integrated delivery models depends on a wide range of resources, [23, 35, 36]. And with the massive cuts in funding of the HIV vertical program, integrating ART just like other HIV-related services may be a novel path. Whether or not policymakers decide to integrate ART services at the departmental level, their success will be largely determined by the system and resources available.

## Limitations

Our study had several limitations. First, the few selected health facilities and the number of healthcare providers interviewed within them may not be representative of the reference population, therefore restraining the generalizability of our findings. We are unable to determine whether ART service integration at departmental levels can have a positive effect on the long-term outcomes of HIV-related care. Thirdly, our study did not include lower-level health facilities that are providing ART services, and that could limit the generalizability of our findings.

## Conclusion and recommendations

Our study shows that the level of readiness for integration of ART services at departmental levels is high among primary healthcare providers. The care providers, furthermore, were confident about the model of care for HIV patients that integrates ART services into departments in resource-limited settings. Policymakers should consider exploring the integration of ART services into departmental levels as a novel path to providing care while maintaining success given the reduction in funding for vertical HIV care. We recommend that future studies be conducted to determine the acceptability of the ART-integrated model of care. Health systems need to invest in training health workers to provide comprehensive ART services and ensure that adequate medical supplies and drugs are available. Furthermore, to confirm our findings, a cluster randomized controlled trial and assessment of the cost-effectiveness analysis, quality of care, and long-term impact of the integration on health outcomes should be conducted.

## Supporting information

**S1 Data.**
(DTA)

## Acknowledgments

We thank the study participants for taking the time to participate in this research. We also acknowledge the efforts of Associate Professor Edward Kumakech, the Co-Investigator at Lira University, and Ms. Guti Jastine for the support they provided during this research project. We are grateful to Dr. Kabunga Amir for extensively reviewing this manuscript. Last, we thank Dr. Ocan Moses for his technical input during the analysis and drafting of the manuscript.

## Author Contributions

**Conceptualization:** Emmanuel Asher Ikwara, Lakeri Nakero, Rogers Isabirye, Syliviah Namutebi, Godfrey Mwesiga, Sean Steven Puleh.

**Data curation:** Emmanuel Asher Ikwara, Lakeri Nakero, Maxson Kenneth Anyolitho, Rogers Isabirye, Syliviah Namutebi, Godfrey Mwesiga, Sean Steven Puleh.

**Formal analysis:** Lakeri Nakero, Syliviah Namutebi, Godfrey Mwesiga, Sean Steven Puleh.

**Funding acquisition:** Emmanuel Asher Ikwara, Rogers Isabirye, Syliviah Namutebi, Godfrey Mwesiga, Sean Steven Puleh.

**Investigation:** Emmanuel Asher Ikwara, Lakeri Nakero, Sean Steven Puleh.

**Methodology:** Emmanuel Asher Ikwara, Sean Steven Puleh.

**Visualization:** Maxson Kenneth Anyolitho.

**Writing – original draft:** Emmanuel Asher Ikwara, Lakeri Nakero, Maxson Kenneth Anyolitho, Rogers Isabirye, Syliviah Namutebi, Godfrey Mwesiga, Sean Steven Puleh.

**Writing – review & editing:** Emmanuel Asher Ikwara, Lakeri Nakero, Maxson Kenneth Anyolitho, Rogers Isabirye, Syliviah Namutebi, Godfrey Mwesiga, Sean Steven Puleh.

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
