## [Decision Letter · Decision Letter 0]

16 Jan 2023

PONE-D-22-30349Determinants of primary healthcare providers’ readiness for integration of ART services in departmental levels: A case study of Lira City and District, UgandaPLOS ONE

Dear Dr. Puleh,

Thank you for submitting your manuscript to PLOS ONE. After careful consideration, we feel that it has merit but does not fully meet PLOS ONE’s publication criteria as it currently stands. Therefore, we invite you to submit a revised version of the manuscript that addresses the points raised during the review process.

Please note that we have only been able to secure a single reviewer to assess your manuscript. We are issuing a decision on your manuscript at this point to prevent further delays in the evaluation of your manuscript. Please be aware that the editor who handles your revised manuscript might find it necessary to invite additional reviewers to assess this work once the revised manuscript is submitted. However, we will aim to proceed on the basis of this single review if possible. 

We look forward to receiving your revised manuscript.

Kind regards,

Steve Zimmerman, PhD

Associate Editor, PLOS ONE

Journal Requirements:

Reviewers' comments:

Reviewer's Responses to Questions

**Comments to the Author**

1. Is the manuscript technically sound, and do the data support the conclusions?

Reviewer #1: Yes

2. Has the statistical analysis been performed appropriately and rigorously? 

Reviewer #1: Yes

3. Have the authors made all data underlying the findings in their manuscript fully available?

Reviewer #1: Yes

4. Is the manuscript presented in an intelligible fashion and written in standard English?

Reviewer #1: Yes

5. Review Comments to the Author

Reviewer #1: Determinants of primary healthcare providers’ readiness for integration of ART services in departmental levels: A case study of Lira City and District, Uganda

I was very interested to review this manuscript and found relevant as far as the health system is concerned in today’s resource context. Here, I’ve highlighted my comments to be seen by authors for the improvement of this manuscript before publication.

Abstract

• Results: It would be better if you incorporate the 95% confidence interval for the level of readiness for integration of ART services at departmental levels.

• Conclusions and recommendations:

o Avoid the term ‘recommendations’ since the conclusions are meant to also consider the potential recommendations.

o Conclusions for relevant determinants not included

Introduction

• The narration “The current success in the effectiveness of the vertical programs has been attributed to the donor funds [4]. In Uganda and across the region, HIV services are provided by vertically operating separately from other health system functions [5]. Notwithstanding the achievements and triumphs already realized, several questions around its sustainability are becoming very critical.” (Page 3 line numbers 73-77) is a duplication of description (page 3 line numbers 57-61). Please reconsider it.

• The authors didn’t mention the potential determinants of providers’ readiness to service integration approach identified by previous studies. It seems only a baseline study of the magnitude or level of readiness to such integration issue. The authors should incorporate such point somewhere here in the introduction section. Otherwise, avoid the determinants in the whole manuscript including the title.

• What is known about the level of integration as well as the determinants should be highlighted clearly.

Materials and methods

• Primary healthcare providers versus the health facilities. Which one was the focus of this study? (Page 4 line numbers 98-102)

• The qualitative part lacked important details for example data analysis software, how trustworthiness managed, who collected data and how many themes developed.

Discussion: It would be better to add more similar empirical evidence in the literature to make the discussion more appropriate.

6. PLOS authors have the option to publish the peer review history of their article (what does this mean?). If published, this will include your full peer review and any attached files.

Reviewer #1: No

---

## [Author Response · Author response to Decision Letter 0]

26 Feb 2023

Dear Editor,

We wish to resubmit our revised journal article titled the Determinants of primary healthcare providers’ readiness for integration of ART services in departmental levels: A case study of Lira City and District, Uganda. All the supporting files requested has been attached. 

We thank you in advance.

Sean Steven Puleh

REVIEWERS COMMENTS RESPONSES LINE NUMBER

Abstract 

• Results: It would be better if you incorporate the 95% confidence interval for the level of readiness for integration of ART services at departmental levels. Thank you highlighting this gap. We have addressed it by inserting the 95% confidence interval as advised. Readiness was 75.2% (95% CI; 0.703 - 0.795) Line 29

• Conclusions and recommendations: 

o Avoid the term ‘recommendations’ since the conclusions are meant to also consider the potential recommendations. We are grateful you pointed out this and it has been deleted as advised Line 38 

o Conclusions for relevant determinants not included

 We have adjusted the conclusion by adding the necessary determinants. The PHCPs who were knowledgeable and those who had worked for at least 6 months at the current facility were strong determinants of the integration of ART services in resource-limited settings. Lines 39 - 41

Introduction 

The narration “The current success in the effectiveness of the vertical programs has been attributed to the donor funds [4]. In Uganda and across the region, HIV services are provided by vertically operating separately from other health system functions [5]. Notwithstanding the achievements and triumphs already realized, several questions around its sustainability are becoming very critical.” (Page 3 line numbers 73-77) is a duplication of description (page 3 line numbers 57-61). Please reconsider it. Thank you for detecting the duplication in the write up. We have made the necessary adjustment in the write up Line 73 - 77

The authors didn’t mention the potential determinants of providers’ readiness to service integration approach identified by previous studies. It seems only a baseline study of the magnitude or level of readiness to such integration issue. The authors should incorporate such point somewhere here in the introduction section. Otherwise, avoid the determinants in the whole manuscript including the title. Thank you for raising an important part of the introduction. Some of the potential determinants include; the current ART service delivery models, health worker capacity, infrastructure, and financing mechanisms. Lines 75 - 79

Materials and methods

• Primary healthcare providers versus the health facilities. Which one was the focus of this study? (Page 4 line numbers 98-102)

 Thank you for highlighting this error, the focus of the study is on the primary healthcare providers. Line 103

• The qualitative part lacked important details for example data analysis software, how trustworthiness managed, who collected data and how many themes developed. Thank you for raising this important aspect of the methodology. We have tried to the best of our ability to address as requested. The data was collected by three research assistants of social science background. 

The software used was Nvivo and how the trustworthiness managed has been described. Lines 140 - 167

Discussion: 

It would be better to add more similar empirical evidence in the literature to make the discussion more appropriate. Much as we appreciate this concern, we were able to get a limited number of empirical data which helped us discuss our findings better. Lines 434 - 501

---

## [Editor Report · Decision Letter 1]

19 Sep 2023

PONE-D-22-30349R1Determinants of primary healthcare providers’ readiness for integration of ART services at departmental levels: A case study of Lira City and District, UgandaPLOS ONE

Dear Dr. Puleh,

Thank you for submitting your manuscript to PLOS ONE. After careful consideration, we feel that it has merit but does not fully meet PLOS ONE’s publication criteria as it currently stands. Therefore, we invite you to submit a revised version of the manuscript that addresses the points raised during the review process.

We look forward to receiving your revised manuscript.

Kind regards,

Sarah Nanzigu, Ph.D.,MSc.,MBchB

Academic Editor

PLOS ONE

Journal Requirements:

**Additional Editor Comments:**

In order to proceed with your manuscript, it is recommended that you make timely and accurate corrections with respect to critical points raised below:

1. Re-write the 'recommendations for relevant determinants' in your abstract and make the information consistent with what you presented in your response letter.

Statement in the response letter: PHCPs who were knowledgeable and those who had worked for at least 6 months (perhaps you wanted to write 6 years) at the current facility, were strong determinants for the

integration of ART services in resource limited settings. Lines 39-41

Statement in the abstract: The PHCPs who were knowledgeable are key at achieving integration of ART services at departmental level. Most of the respondents who had worked for at at least 6 months (perhaps you

wanted to write 6 years) at the current facility were strong determinants of the integration of ART services in resource limited settings. Lines 39-41

Also, be specific with what 'knowledge' you are referring to, and what you mean with 'current health facility'. If the 'PHCPs' awareness or knowledge on the integration of ART services', was a determinant, be

specific in your communication. Likewise, a decade from now, readers may not understand your term 'current health facility', so be explicit with your description of 'current'.

2. The information on recommendations in the revised abstract is passive, erroneously written, and has lost the impact of a recommendation. Lines 42-46 reads " We recommend that policymakers should consider exploring training the healthworkers to enhance improving the integration of ART services into departmental levels as a novel path. be conducted to determine the long-term impact of the integration on health outcomes using cluster randomized controlled trials". Please read your detailed recommendations in lines 515-523, and summarize precisely the recommendations in your abstract.

3. Attend to grammatical and typographic errors in the following statements

- Lines 158-159: . A seven-step thematic analysis model by Clerke and Braun was used to analyse qualitative data was analyzed using thematic analysis [21].

- Lines 162: over 15 themes

- Lines 457-460: Our findings are consistent with evidence from elsewhere, which found that integrating HIV services are increased patient satisfaction, increased perceived quality of care, and increased patient

access to services and patient health outcomes [24, 25].

- Lines 542-543: Ikwara Emmanuel Asher designed conceptualized, designed the study and designed the study instruments, obtained the grant and necessary approvals.

- Lines 546-548: Sean Steven Puleh conceptualized and designed the study, designed the study instruments, led data collection activities, and substantially contributed to paper development. in addition to providing

mentorship during study conduct.

---

## [Author Response · Author response to Decision Letter 1]

21 Sep 2023

Dear Editor,

We wish to resubmit our revised journal article titled the Determinants of primary healthcare providers’ readiness for integration of ART services in departmental levels: A case study of Lira City and District, Uganda. All the supporting files requested has been attached. 

We thank you in advance.

Sean Steven Puleh

---

## [Editor Report · Decision Letter 2]

25 Sep 2023

Determinants of primary healthcare providers’ readiness for integration of ART services at departmental levels: A case study of Lira City and District, Uganda

PONE-D-22-30349R2

Dear Dr. Puleh,

We’re pleased to inform you that your manuscript has been judged scientifically suitable for publication and will be formally accepted for publication once it meets all outstanding technical requirements.

Kind regards,

Sarah Nanzigu, Ph.D.,MSc.,MBchB

Academic Editor

PLOS ONE

Additional Editor Comments (optional):

Please review line 40-42: You concluded that "Notably, PHCs knowledge about integration and those who spent at least six months at the current health facility of work, were strong determinants for the integration of ART services in resource limited settings." The period mentioned in this statement 'six months' is not in agreement with the period (6 years) provided in the results section (lines 427-428, and 432); Moreover, your methods section indicates that you purposively recruited only participants who had work experience of at least 6 months (line 129), which means there were no comparative counterparts with working experience below 6 months.

During the previous editor's comments, a hint was given to look at this period (6months) as a possible error. We once again invite you to affirm or correct the above statement in all your abstracts.

Otherwise, an erratum is always recommended for errors discovered after publication.
---

## [Editor Report · Acceptance letter]

28 Sep 2023

PONE-D-22-30349R2 

Determinants of primary healthcare providers’ readiness for integration of ART services at departmental levels: A case study of Lira City and District, Uganda 

Dear Dr. Puleh:

I'm pleased to inform you that your manuscript has been deemed suitable for publication in PLOS ONE. Congratulations! Your manuscript is now with our production department. 

Kind regards, 

on behalf of

Dr. Sarah Nanzigu 

Academic Editor

PLOS ONE